# Analysis of *Trypanosoma equiperdum* Recombinant Proteins for the Serological Diagnosis of Dourine

**DOI:** 10.3390/vetsci11030127

**Published:** 2024-03-13

**Authors:** Mirella Luciani, Gisella Armillotta, Tiziana Di Febo, Ivanka Krasteva, Simonetta Ulisse, Chiara Di Pancrazio, Caterina Laguardia, Fabrizia Perletta, Anna Serroni, Marta Maggetti, Lilia Testa, Flavio Sacchini, Mariangela Iorio, Diamante Rodomonti, Manuela Tittarelli, Maria Teresa Mercante

**Affiliations:** Istituto Zooprofilattico Sperimentale dell’Abruzzo e del Molise, Via Campo Boario, 64100 Teramo, Italy; m.luciani@izs.it (M.L.); g.armillotta@izs.it (G.A.); s.ulisse@izs.it (S.U.); c.dipancrazio@izs.it (C.D.P.); c.laguardia@izs.it (C.L.); f.perletta@izs.it (F.P.); a.serroni@izs.it (A.S.); m.maggetti@izs.it (M.M.); l.testa@izs.it (L.T.); f.sacchini@izs.it (F.S.); m.iorio@izs.it (M.I.); d.rodomonti@izs.it (D.R.); m.tittarelli@izs.it (M.T.); t.mercante@izs.it (M.T.M.)

**Keywords:** dourine, ELISA, immunoblotting, recombinant proteins

## Abstract

**Simple Summary:**

Dourine is a contagious disease prevalent in equids that can lead to serious health issues in affected animals, including reproductive problems and nervous system disorders. Stamping out policies implemented during the 20th century has been crucial in reducing the prevalence of dourine. To maintain control and prevent the international spread of dourine, stringent measures are necessary when it comes to the movement of equids across borders. The challenges associated with diagnosing this disease, including its coexistence with closely related trypanosomes, highlight the critical need for accurate and specific diagnostic tools. In this study, three proteins, namely peptidyl-prolyl cis-trans isomerase (A0A1G4I8N3), GrpE protein homolog (A0A1G4I464), and transport protein particle (TRAPP) component, putative (A0A1G4I740), were expressed as recombinant proteins. In a previous study based on bioinformatics analyses, these proteins were unique to *T. equiperdum* and did not exhibit matches or cross-reactivities with other *Trypanosoma* species protein; moreover, these proteins were found to be immunogenic because they exhibit the presence of potential B-cell epitopes. The three proteins were screened for reactivity to a small panel of horse sera using indirect enzyme-linked immunosorbent assay and an immunoblotting test. Protein A0A1G4I8N3, giving the best results, will be tested with a larger amount of sera to more comprehensively evaluate its potential use in the diagnosis of dourine.

**Abstract:**

The significance of *Trypanosoma equiperdum* as the causative agent of dourine cannot be understated, especially given its high mortality rate among equids. International movement of equids should be subject to thorough health checks and screenings to ensure that animals are not infected with *Trypanosoma equiperdum*. This involves the implementation of quarantine protocols, testing procedures, and the issuance of health certificates to certify the health status of the animals. Three proteins, the peptidyl-prolyl cis-trans isomerase (A0A1G4I8N3), the GrpE protein homolog (A0A1G4I464) and the transport protein particle (TRAPP) component, putative (A0A1G4I740) (UniProt accession numbers SCU68469.1, SCU66661.1 and SCU67727.1), were identified as unique to *T. equiperdum* by bioinformatics analysis. The proteins were expressed as recombinant proteins and tested using an indirect ELISA and immunoblotting test with a panel of horse positive and negative sera for dourine. The diagnostic sensitivity, specificity and accuracy of the i-ELISAs were 86.7%, 53.8% and 59.0% for A0A1G4I8N3; 53.3%, 58.7% and 57.9% for A0A1G4I464; and 73.3%, 65.0% and 66.3% for A0A1G4I740, respectively, while the diagnostic sensitivity, specificity and accuracy of immunoblotting were 86.7%, 92.5% and 91.6% for A0A1G4I8N3; 46.7%, 81.3% and 75.8% for A0A1G4I464; and 80.0%, 63.8% and 66.3% for A0A1G4I740. Among the three proteins evaluated in the present work, A0A1G4I8N3 provided the best results when tested by immunoblotting; diagnostic application of this protein should be further investigated using a greater number of positive and negative sera.

## 1. Introduction

*Trypanosoma equiperdum* is the causative agent of dourine, a chronic or acute contagious disease prevalent in equids. Symptoms of the disease include fever, skin lesions, edema, weight loss, and general weakness, while some horses may develop muscular paralysis. The disease is often fatal. Currently, there is no specific cure for dourine. Control measures involve surveillance, quarantine and culling of infected animals due to the highly contagious nature of the disease. Dourine is the only sexually transmitted trypanosomiasis and does not involve invertebrate vectors. Moreover, *T. equiperdum* differs from other trypanosomes since it is mainly detected in the host tissues and only occasionally in the blood. There are no known natural reservoirs of the parasite other than infected equids [1]. The mortality rate of dourine exceeds 50% [2,3].

*T. equiperdum* is morphologically identical to other insect vector-transmitted parasites such as *Trypanosoma evansi* and *Trypanosoma brucei*, which cause surra and nagana, respectively. In many regions of the world, these three parasite species occur together, and current diagnostic tests are unable to differentiate among them [3,4]. Recently, according to phylogenetic analysis, some authors have suggested that *T. equiperdum* and *T. evansi* may have evolved from *T. brucei* and should be classified as subspecies (*T. brucei equiperdum* and *T. brucei evansi*) [5,6].

The diagnosis of dourine can be challenging due to limited understanding of the pathogenesis of the disease. The diagnostic process usually relies on observing the animals’s medical history and clinical symptoms, followed by pathological findings and serological tests, such as a complement fixation test (CFT) [7].

Antibodies are present in infected animals regardless of whether they show clinical signs or not. Therefore, different serological methods have been developed to diagnose infection. CFTs are frequently used as the primary test to investigate suspected cases with clinical signs or latent carriers. However, some non-infected animals, especially donkeys, often produce results that are difficult to interpret due to the anticomplementary effects of the sera tested. Hence, the indirect fluorescent antibody test (IFAT) based on IgG is used as a confirmatory test, while an indirect ELISA is also applied for diagnosis. All these serological methods rely on whole *T. equiperdum* antigen, typically produced in experimental animals (rats) [1]. In fact, recombinant proteins of *T. equiperdum* capable of replacing the whole antigen are currently not available. Moreover, the complete genome of *T. equiperdum* was only published in early 2017 [8], and to date, literature on the production and diagnostic use of recombinant *T. equiperdum* proteins is scarce [9].

Similarly, genetic markers that unequivocally differentiate *T. equiperdum* from *T. evansi* within the subgenus Trypanozoon have not been identified, and so, consequently a *T. equiperdum*-specific polymerase chain reaction (PCR) method is lacking [1].

Dourine serodiagnosis should be improved using selected *T. equiperdum* recombinant proteins and monoclonal antibodies.

In previous work [10] a chemiluminescent immunoblotting assay (cIB) was developed and used to study *T. equiperdum* antigen patterns recognised by serum antibodies from uninfected and infected animals. Results revealed that antibodies from infected horses specifically bind to *T. equiperdum* due to low molecular weight bands ranging from 16 to 35 kDa, in contrast to antibodies from healthy horses that recognise only bands with molecular weights higher than 37 kDa. In a subsequent work [11], immunoreactive proteins contained in these bands were identified by mass spectrometry and bioinformatics analysis. Three of these proteins, the peptidyl-prolyl cis-trans isomerase (A0A1G4I8N3), the GrpE protein homolog (A0A1G4I464), and the transport protein particle (TRAPP) component, putative (A0A1G4I740) (UniProt accession numbers SCU68469.1, SCU66661.1 and SCU67727.1), were found to be unique for *T. equiperdum* as they do not exhibit matches with other *Trypanosoma* species proteins by bioinformatics analysis; moreover, they were proven immunogenic because they exhibit the presence of potential B-cell epitopes. 

The aim of this study was to produce A0A1G4I8N3, A0A1G4I464 and A0A1G4I740 as recombinant proteins and to evaluate their diagnostic potential in indirect ELISAs and immunoblotting assays.

## 2. Materials and Methods

### 2.1. Expression and Production of T. equiperdum Recombinant Proteins

Small scale production of the 3 recombinant proteins was performed by GenScript (GenScript, Piscataway, NJ, USA Inc.). Proteins A0A1G4I8N3 and A0A1G4I464 were expressed in baculovirus using cloning vector pFastBac1 in a Sf-900II SFM (1X) medium culture, while protein A0A1G4I740 was expressed in *Escherichia coli* using cloning vector pET-30a(+) in Terrific broth. Recombinant proteins were designed on the full-length proteins and were fused to an N-terminal His6 (Histidine6) tag. The amino acid sequence (FASTA format) and cloning strategy of the three proteins are shown, respectively, in Table 1 and Table 2. 

The three proteins were purified by one-step purification using Ni-NTA columns; A0A1G4I8N3 and A0A1G4I464 were obtained from the supernatant of the cell lysate, while the protein A0A1G4I740 was obtained from inclusion bodies. Purified proteins A0A1G4I8N3 and A0A1G4I464 were stored at −80 °C in 50 mM Tris-HCl, 500 mM NaCl, 5% glycerol, pH 8.0, while the purified protein A0A1G4I740 was stored at −80 °C in 50 mM Tris-HCl, 150 mM NaCl, 10% Glycerol, 1 M L-Arginine, pH 8.0.

Large scale production of A0A1G4I8N3 and A0A1G4I464 was performed at Istituto Zooprofilattico Sperimentale dell’Abruzzo e del Molise “G. Caporale” (IZSAM). The P2 viral stock, supplied by GenScript for the two proteins, was amplified in order to obtain the P3 viral stock. Erlenmeyer flasks (Corning, Corning, NY, USA) containing 100 mL of suspension of Sf9 insect cell line (ECACC 05011001) at a cell density of 1.6 × 10^6^ cells/mL (Countess, Invitrogen, Waltham, MA, USA) in Sf-900 II SFM culture medium (Gibco, Waltham, MA, USA) were added with P2 viral stock at a multiplicity of infection (MOI) of 0.1. The suspension was incubated at 27 °C in a thermo-refrigerated shaker (MaxQTM 6000, Thermo Fisher Scientific, Waltham, MA, USA) at a speed of 110 rpm. The viral suspensions were harvested at 48 h post infection (p.i.), and at 72 h p.i. P3 was obtained by centrifugation at 3000× *g* for 30 min for the two separately collected viral suspensions. Titrations were performed on the P3 products using the limit dilution method with Sf9ET insect cells (Easy Titer) [12]. The P3 viral stock collected at 72 h p.i. showed a higher titre than the one collected at 48 h.

Before proceeding with the production of recombinant proteins on a large scale, it was necessary to carry out preliminary tests in order to identify the optimal production parameters.

Therefore, small productions were prepared using an intermediate volume between small and large scale. The parameters taken into consideration were the Multiplicity of infection (MOI), the time of infection, coinciding with the cell density (TOI), and the time of collection (TOH). Erlenmeyer flasks with a capacity of 1 L were used with 400 mL of cell suspension at an initial cell density of 3.5 × 10^6^ cells/mL incubated at a temperature of 27 °C in a heat-refrigerated shaker at 110 rpm. The P3 viral stock used for the production of both recombinant proteins was the one collected at 72 h p.i.

For each recombinant protein, six flasks were prepared, three of which were infected with 0.1 MOI and three with 0.01 MOI, while their cells were harvested at 48, 72 and 96 h p.i.

The large-scale production was carried out using the parameters resulting from the tests described above. For this purpose, 1500 mL of Sf9 cell cultures at an initial cell density of 3.5 × 10^6^ cells/mL were infected with P3 at 0.01 MOI, incubated at 27 °C, and harvested at 72 h p.i. 

The collected suspension of each protein was centrifuged at 3000× *g* for 10 min, while the supernatants were purified using an affinity chromatographic column (IMAC). The purified proteins were resuspended in 0.01 M PBS, pH 7.2, containing 0.05% Sarkosyl, and the protein concentration was evaluated using the Bradford assay. Both proteins were stored at 4 °C until use.

### 2.2. Serum Panel

Fifteen positive sera were collected from naturally infected horses during the Italian outbreak in 2011 and during the African outbreaks (Namibia); African sera were supplied by the Namibian Central Veterinary Laboratory (CVL, Windhoek, Namibia). All sera were positive to CFT (titers ranging from 1:10 to 1:640) and confirmed positive by an IFAT. Sera from 80 Italian healthy animals were collected during the national surveillance plans for equine diseases in Italy. Moreover, three sera positive for *Theileria equi* and three sera positive for *Babesia caballi*, stored in the IZSAM serum bank, were also used to test cross-reactions of ELISAs and imunoblotting with other protozoans infecting horses. As positive control, a serum collected from a naturally infected horse (batch 019/1994, CFT titre 1:320) was used.

### 2.3. Indirect ELISA

Preliminary experiments were carried out to optimise the ELISA protocol in terms of the type of microplate (Polysorp and Maxisorp, Nunc) and antigen dilution for the coating, blocking buffer composition and serum dilutions. 

Purified recombinant proteins were diluted in 0.05 M carbonate-bicarbonate buffer pH 9.6 to a final concentration of 2.5 μg/mL for A0A1G4I8N3 and A0A1G4I464 and 5 μg/mL for A0A1G4I740, then dispensed (100 μL/well) into 96-well microplates and incubated overnight at +4 °C. Proteins A0A1G4I8N3 and A0A1G4I464 were coated onto Maxisorp microplates (Nunc), and the protein A0A1G4I740 was coated onto Polysorp microplates (Nunc). The plates were then blocked for 1 h at room temperature (RT) with bovine seroalbumin (BSA) diluted in PBS-T (1% BSA for proteins A0A1G4I8N3 and A0A1G4I464; 5% BSA for A0A1G4I740). After three washes with PBS-T, 100 μL/well of sera diluted in PBS-T containing BSA were added for 1 h at RT. The sera were tested in duplicate at the following dilutions: 1:5 in PBS-T containing 1% BSA in the i-ELISAs performed with proteins A0A1G4I8N3 and A0A1G4I464; 1:10 in PBS-T containing 5% BSA in the i-ELISA performed with A0A1G4I740. One-hundred μL/well of serum dilution buffer was added in two wells of each microplate as blank. Plates were then washed and incubated at RT for 30 min (A0A1G4I8N3 and A0A1G4I464 proteins) or 1 h (A0A1G4I740 protein) with 100 μL/well of MAb-HRP 1B10B6E9 anti horse-IgG (IZSAM) [13] diluted 1:100,000 in PBS-T. After further washes, 100 μL of 3-3′-5-5′ tetramethylbenzidine (TMB, Surmodics) were dispensed into each well, and the plates were incubated at RT for 30 min. The reaction was stopped by adding 50 μL/well of 0.5 N sulphuric acid. Optical densities at 450 nm (OD_450_) were normalised using the following formula: S/P% = (mean OD_450_ sample serum/mean OD_450_ Positive Control) × 100.

For each protein, the cut-off value, diagnostic sensitivity, specificity and accuracy of the i-ELISA were calculated using receiver operator characteristic (ROC) [14], while the CFT was considered the gold standard.

### 2.4. Immunoblotting

Recombinant proteins (0.5 μg per well of A0A1G4I8N3, A0A1G4I464 and A0A1G4I740), diluted in SDS-PAGE denaturing buffer (Novex, Life Technologies, Carlsbad, CA, USA), were loaded into NuPAGE^®^ 4–12% Bis-Tris gels (Novex, Life Technologies) and separated at a constant voltage of 200 V. Afterwards, proteins were blotted the on iBlot2^®^ mini nitrocellulose membranes (Novex, Life Technologies) using iBlot2^®^ Dry Blotting System (Novex, Life Technologies) at 20 V for 1 min, 23 V for 4 min and 25 V for 2 min. Membranes were then blocked for 2 h at RT with Roti Block 10x (Carl Roth) diluted 1:10 in deionised water (Roti Block 1x) for the protein A0A1G4I8N3 and A0A1G4I740 and 5% skimmed milk for the protein A0A1G4I464, then incubated overnight at RT with sera diluted 1:5000 and 1:1500 in Roti Block 1x for A0A1G4I8N3 and A0A1G4I740, respectively. For the protein A0A1G4I464, the sera were diluted 1:20 in PBS-T containing 2.5% skimmed milk. After three washes with PBS-T for 10 min, membranes were incubated for 1 h at RT with Mab-HRP 1B10B6E9 anti horse-IgG (IZSAM) [13] diluted 1:100,000 in Roti Block 1x. After three washes with PBS-T, and one final wash with PBS for 10 min, membranes were detected by chemiluminescence (ECL Western blotting detection kit, GE Healthcare, Chicago, IL, USA) using the Chemidoc MP (Bio-Rad, Hercules, CA, USA). Analyses were performed using Image Lab Software version 4.0.1 (Bio-Rad). 

For each protein, diagnostic sensitivity, specificity and accuracy of the immunoblotting test were estimated using the software Microsoft Excel version 2016 and contingency tables; a CFT was considered the gold standard.

## 3. Results

### 3.1. Expression and Production of T. equiperdum Recombinant Proteins

A0A1G4I8N3 and A0A1G4I464 were expressed as recombinant proteins in baculovirus; while the protein A0A1G4I740 was poorly expressed in the baculovirus system, so a change in expression system was necessary. 

Molecular weights of recombinant proteins A0A1G4I8N3, A0A1G4I464 and A0A1G4I740, expressed as His6 fusion proteins, were as follows: 34.5 kDa (28.5 kDa + 6 His), 30.7 kDa (24.7 kDa + 6 His) and 30.3 kDa (24.3 kDa + 6 His), respectively (Figure 1 and Figure 2, The original image is in the Appendix A). The predicted isoelectric points were 10.35 (A0A1G4I8N3), 7.40 (A0A1G4I464) and 7.69 (A0A1G4I740).

The degrees of purity obtained following small scale production were 70% for A0A1G4I8N3 55% for A0A1G4I464 and 90% for A0A1G4I740. The degrees of purity of A0A1G4I8N3 and A0A1G4I464 produced on the large scale were similar to those obtained on the small scale. Moreover, proteins produced in small and large volumes provided comparable SDS-PAGE profiles.

### 3.2. Indirect ELISA

Selected cut-off values for the i-ELISAs performed using A0A1G4I8N3, A0A1G4I464 and A0A1G4I740 proteins were, respectively, 50%, 80% and 55%. The A0A1G4I8N3-i-ELISA provided the following results: 43 out of 80 negative sera saw a negative result and 37 resulted positively, while 13 out of 15 positive sera resulted positively with two negative results. The sensitivity, specificity and accuracy of the test were, respectively, 86.7% (95% L.C. 53.7–96.0%), 53.8% (95% L.C. 42.9–64.3%) and 59.0% (95% L.C. 48.9–68.3%) (Table 3). 

The A0A1G4I464-i-ELISA results were as follows: 47 out of 80 negative sera resulted negatively and 33 resulted positively; 8 out of 15 positive sera resulted positively and 7 resulted negatively. The sensitivity, specificity and accuracy of the test were, respectively 53.3% (95% L.C. 23.6–75.3%), 58.7% (95% L.C. 47.8–68.9%) and 57.9% (95% L.C. 47.8–67.3%) (Table 4). 

The A0A1G4I740-i-ELISA provided the following results: 52 out of 80 negative sera resulted negatively and 28 resulted positively; 11 out of 15 positive sera resulted positively and 4 negatively. The sensitivity, specificity and accuracy of the test were, respectively, 73.3% (95% L.C. 47.6–89.0%), 65.0% (95% L.C. 54.0–74.6%) and 66.3% (95% L.C. 56.3–75.0%) (Table 5). 

For the three proteins, two sera positive for *B. caballi* and 1 positive for *T. equi* resulted positive; 1 serum positive for *B. caballi* and 2 positives for *T. equi* resulted negative.

### 3.3. Immunoblotting

Blotting performed using protein A0A1G4I8N3 provided the following results: 74 out of 80 negative sera resulted negatively, while 13 out of 15 positive sera resulted positively. The sensitivity, specificity and accuracy of the test, calculated for the examined sera, were, respectively, 86.7% (95% L.C. 61.7–96.0%), 92.5% (95% L.C. 84.6–96.5%) and 91.6% (95% L.C. 84.2–95.6%) (Table 6). 

Regarding the recombinant protein A0A1G4I464, 7 out of 15 positive sera resulted positively and 8 negatively; 65 out of 80 negative sera resulted negatively and 15 positively. Therefore the sensitivity, specificity and accuracy of the test were, respectively 46.7% (95% L.C. 24.7–70.1%), 81.3% (95% L.C. 71.3–88.3%) and 75.8% (95% L.C. 66.3–83.3%) (Table 7).

For the protein A0A1G4I740, 12 out of 15 positive sera resulted positively and 3 negatively; 51 out of 80 negative sera resulted negatively and 29 positively. Therefore the sensitivity, specificity and accuracy of the test were, respectively 80.0% (95% L.C. 54.4–92.7%), 63.8% (95% L.C. 52.8–73.4%) and 66.3% (95% L.C. 56.3–75.0%) (Table 8).

No evidence of cross-reactivity of *B. caballi* and *T. equi* positive panel sera with the three proteins was observed.

## 4. Discussion

Dourine is endemic in many areas of Asia, Africa, Russia, the Middle East and Eastern Europe [1]. Surra is endemic in Asia, North Africa and Central and South America [15]. In 2011, outbreaks of dourine occurred in Italy, almost 10 years after its last appearance [16]; in July 2006 an outbreak of surra, caused by *T. evansi*, was detected in France in camels imported from the Canary Islands [17]. These outbreaks underline the risk of introduction of the trypanosome infection in non-endemic countries. Current serological tests applied for the serological diagnosis of dourine, such as CFTs, IFATs and ELISAs, are carried out using the whole *Trypanosoma* antigen produced in rats and polyclonal secondary antibodies, and they suffer from limited specificity. In the same way, genetic markers that unequivocally differentiate *T. equiperdum* from *T. evansi* within the subgenus Trypanozoon are lacking; therefore development of a *T. equiperdum*-specific PCR remains imperative [1].

The use of recombinant proteins instead of whole antigens could improve serological diagnosis and allow for the development of more specific and sensitive diagnostic tests, in addition to reducing the employment of live animals to be sacrificed for antigen production.

For instance, recombinant proteins from *T. cruzi* are have been proven to offer higher sensitivity and specificity compared to tests using crude antigens [18,19,20,21,22].

The aim of the present study was to test, through an i-ELISA and WB, three selected *T. equiperdum* proteins produced as recombinant antigens in order to evaluate their diagnostic performances in the serological diagnosis of dourine in equids. According to bioinformatic analyses, proteins A0A1G4I8N3, A0A1G4I464 and A0A1G4I740 showed the presence of potential B-cell epitopes in a fraction higher than or equal to 30%, providing results unique to *T. equiperdum* [11]. 

The protein peptidyl-prolyl cis-trans isomerase (A0A1G4I8N3) belongs to the PPIase family. PPIases catalyse the cis/trans isomerization of peptide bonds preceding a prolyl residue in polypeptides during protein folding or refolding after transport of proteins into organelles by stabilising the cis/trans transition state [23]. It was be found that PPIases are involved in several biological processes, such as gene expression, signal transduction, protein secretion, development, and tissue regeneration. Moreover, PPIases were identified as virulence/associated proteins in bacteria and protozoa. The contribution to virulence is variable and dependent on the pleiotropic roles of a single PPIase in the respective pathogen [24].

The protein A0A1G4I464 is an essential component of the PAM complex, required for the translocation of transit peptide-containing proteins from the inner membrane into the mitochondrial matrix in an ATP-dependent manner; it is also involved in protein folding [25].

The protein A0A1G4I740 is a component of Transport protein particle (TRAPP), a highly conserved modular multi-subunit protein complex. Originally identified as a “transport protein particle” with a role in endoplasmic reticulum-to-Golgi transport, its multiple subunits and their conservation from yeast to humans were characterised in the late 1990s. Later it was shown to act as a Ypt/Rab GTPase nucleotide exchanger, GEF, in the 2000s, and to be conserved from yeast to humans, where Rabs are relevant to a wide array of diseases. Trafficking among intracellular compartments is mediated by vesicles and regulated by the highly conserved Ypt/Rab GTPases, their nucleotide exchangers, GEFs, and their downstream effectors. Multi-protein complexes facilitate vesicle formation at donor compartments, as well as subsequent targeting, tethering and fusion with acceptor compartments. Ypt/Rab GTPase act as molecular switches that attach to membranes via lipid tails to recruit their multiple downstream effectors, which mediate vesicular transport [26,27]. 

The performances of i-ELISAs developed using proteins A0A1G4I8N3, A0A1G4I464 and A0A1G4I740 were not satisfying (sensitivity and specificity below 90%); this is due to cross-reactions observed among the three proteins and antibodies in some of the negative sera tested, which cause a decrease in the diagnostic specificity and accuracy values of ELISAs.

During the standardization of ELISAs, various blocking agents such as skim milk, yeast, bovine serum albumin, casein, ovalbumin, bovine and mouse serum were evaluated at concentrations ranging from 0.5% to 10% for use in both the blocking buffer and serum dilution buffer. The goal was to minimize non-specific binding of antibodies present in negative sera. However, no improvement in diagnostic sensitivity, specificity, or accuracy was observed with any of the tested blocking agents and concentrations. Moreover, according to Leibly et. al [28], denaturing agents (dithiothreitol and urea) were added into the serum dilution buffer at low concentrations (0.5 M, 1 M) in order to reduce aspecific reactions of antibodies; however, no improvement in ELISA results, was observed in this case.

Immunoblotting provided better results in terms of diagnostic specificity and accuracy with proteins A0A1G4I8N3 and A0A1G4I464 compared to i-ELISA. In fact, specificity and accuracy for immunoblotting were, respectively, 92.5% and 91.6% for A0A1G4I8N3 and 81.3% and 75.8% for A0A1G4I464; specificy and accuracy for i-ELISA were, respectively, 53.8% and 59.0% for A0A1G4I8N3 and 58.7% and 57.9% for A0A1G4I464. Diagnostic sensitivity was similar for the i-ELISA and immunoblotting (86.7%) for A0A1G4I8N3; for A0A1G4I464, sensitivity of the i-ELISA was a little better than for immunoblotting (53.3% vs. 46.7%). Results for A0A1G4I740 were similar for the i-ELISA and blotting. For the i-ELISA, a colorimetric substrate (tetramethylbenzidine) was used. For immunoblotting, the application of a chemiluminescent substrate allowed to work with higher dilutions of sera (1:5000 for A0A1G4I8N3; 1:20 for A0A1G4I464; 1:1500 for A0A1G4I740) compared with the i-ELISA (1:5 for A0A1G4I8N3 and A0A1G4I464; 1:10 for A0A1G4I740) resulted in a reduction in cross-reactions among unspecific antibodies and the three recombinant proteins used as antigens. Nevertheless, higher dilutions of sera would decrease the diagnostic sensitivity of tests by increasing the number of false negative results, particularly when weak positive sera were tested, as observed for protein A0A1G4I464.

## 5. Conclusions

In conclusion, among the three proteins selected in the present work, A0A1G4I8N3 provided the best results when tested by immunoblotting. Diagnostic application of this protein should be further investigated by testing a higher number of sera. Regarding use of i-ELISAs, performances of this technique may be improved by utilising chemiluminescent substrates [29]; however, this requires suitable microplate readers.

Proteins A0A1G4I8N3, A0A1G4I464 and A0A1G4I740 were found to be unique to *T. equiperdum* after a comparison with all sequenced proteins of other *Trypanosoma* species by bioinformatics analysis. However, predictive information of the characteristics of the proteins obtained by “in silico” analysis is not always confirmed by laboratory tests. One limit of the study was the small number of dourine positive sera available and the lack of sera positive for surra, which would have been useful to assess the *T. equiperdum*-specificity of the recombinant antigens tested. From 2011 to the present, no other outbreaks of dourine have been identified in Europe; therefore, it has not been possible to obtain sera from naturally infected animals in our country. Additionally, for ethical reasons, we chose not to conduct experimental infections to obtain the necessary materials for the study. The genome sequence of *T. equiperdum* has only been available since the beginning of the current year [8]; therefore, to date, there are only a few published works about the diagnostic use of *T. equiperdum* proteins.

The three proteins were selected among 37 proteins found to be unique to *T. equiperdum*; more proteins of *T. equiperdum* should be expressed as recombinant antigens and evaluated by serological tests in order to select those showing optimal performances to improve any future diagnosing of dourine. Availability of recombinant antigens would also be a benefit regarding reducing the use of laboratory animals, in particular rats, for the production of the whole antigen, a method currently employed for diagnostic tests (CFT, IFAT, ELISA) (this is in accordance with the principles of the 3Rs (replacement, reduction and refinement) on the use of animals for scientific purposes) [30].

## Figures and Tables

**Figure 1 vetsci-11-00127-f001:**
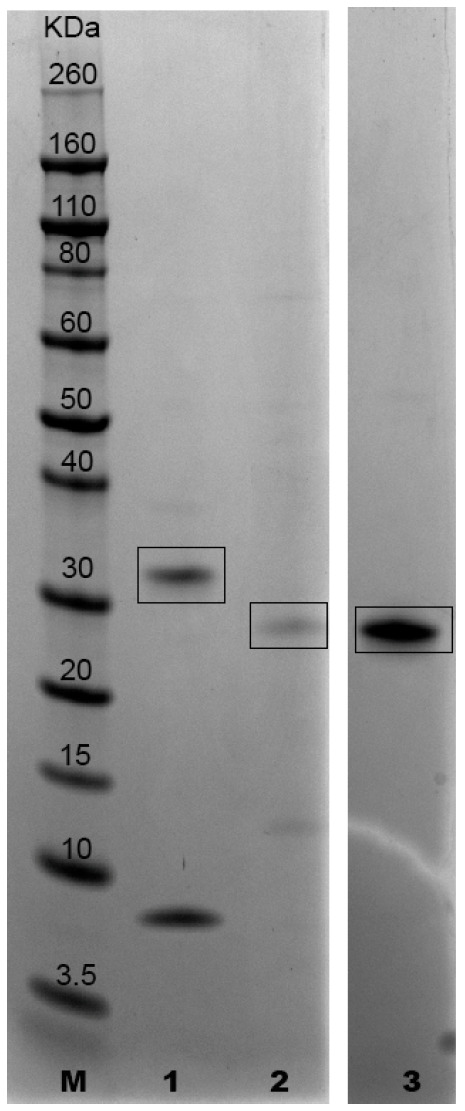
SDS-PAGE and Coomassie stain of purified proteins A0A1G4I8N3, A0A1G4I464 and A0A1G4I740 (GenScript) Lane M: molecular weight standard (Novex Sharp Prestained Protein Standard, Life Technologies); lane 1: protein A0A1G4I8N3 (the band of 9 kDA is a fragment of the protein A0A1G4I8N3 containing the His-tag as declared by the analysis report provided by GenScript); lane 2: protein A0A1G4I464; lane 3: protein A0A1G4I740.

**Figure 2 vetsci-11-00127-f002:**
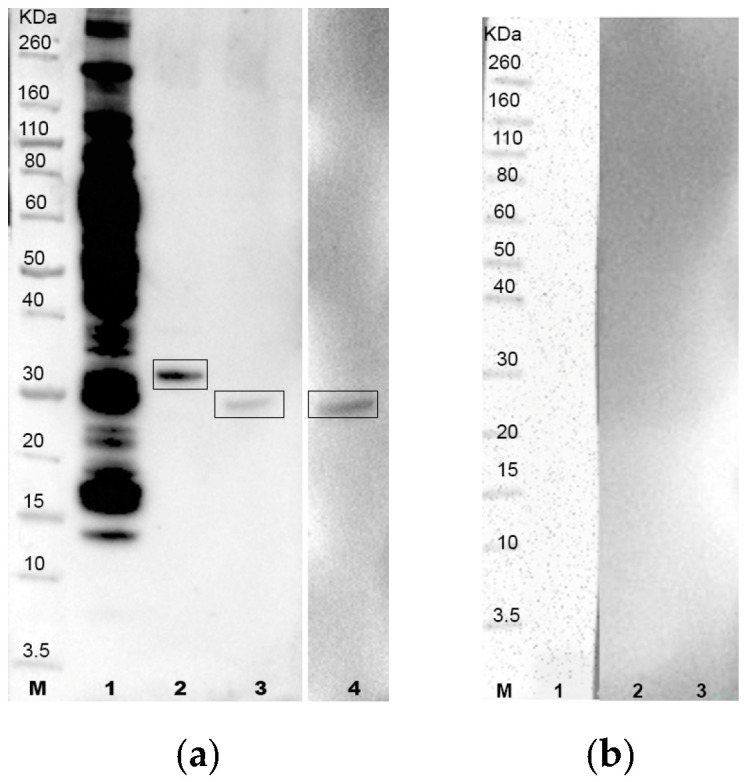
(**a**) Western blotting test using *T. equiperdum* OVI whole antigen (lane 1) and recombinant proteins A0A1G4I8N3 (lane 2), A0A1G4I464 (lane 3) and A0A1G4I740 (lane 4) incubated with the reference horse serum positive for *T. equiperdum* (batch 019/1994, IZSAM); (**b**) *T. equiperdum* OVI whole antigen (lane 1) and recombinant proteins A0A1G4I8N3 (lane 2), A0A1G4I464 (lane 3) and A0A1G4I740 (lane 4) incubated with the reference horse serum negative for *T. equiperdum*. Lane M: molecular weight standard (Novex Sharp Prestained Protein Standard, Life Technologies).

**Table 1 vetsci-11-00127-t001:** Amino acid sequence of proteins A0A1G4I8N3, A0A1G4I464 and A0A1G4I740 (FASTA format).

Protein	Sequence (FASTA)	Protein Length (Aminoacids)
A0A1G4I8N3	MPKVSVPPKTERHGKIEAPETNNPKVFFDVSINEKAAGRIVMELYADTVPRTA ENFRALCTGEKGRGKSGKPLHYKGCIFHRVIPGFMLQGGDITRGNGTGGESIYGTTFR DESFAGKAGKHTGLGCLSMANAGPNTNGSQFFICTANTPWLNGKHVVFGRVTE GIDVVKSIERLGSDSGKTRGRIIIANCGELQTQKEGAAKKEKVKAKEGNNAGKLSTI VEGTGGAKRPREDVEDLEERKKRIREKRERIAKLRAQAEEKHHHHHH	268
A0A1G4I464	MRALTFRSSLCAGRTAVGAMCFGRLWASSTSPTEGSEKQNVTED SETVSVAPVSPEAYAKLEKELSDAKERIAELKKEVLYRAADAENARRIGSEDVTKAKAY GITSFGKDMLDVVDTLERGLEAITKLPQAEVEGHKTLSSIHTGIKLSLKLLLNNLA KHGIEKLDVAVGAKFDPNFHDALLKVPPTAEAPPGHISTVLKTGYKIQDRVLRA SQVGVASDDHHHHHH	228
A0A1G4I740	MSQHGLVSESVVSELSLELVSYALRGNSQKKEINFCRNKEVDTEFG SKGIERLGLLVGLRSAERLLYREATFGGSTPNDVARFVGQHLWKTVFGKKVDRMKH MDKIYFCLIDNNFRWLQGFSDAKSDQTVSAVDGYPYDSSEKYCGGDGPTDQGKESG SVLPPDSDVLRYAVSILRGFVQVMYPSGPIKIQASRNEKGETQFVLDFRSVA THHHHHH	217

**Table 2 vetsci-11-00127-t002:** Cloning strategy of proteins A0A1G4I8N3, A0A1G4I464 and A0A1G4I740.

Protein	Cloning Strategy
A0A1G4I8N3	*Eco*RI-Kozak sequence-A0A1G4I8N3-His—stop codon—*Hind*III
A0A1G4I464	*Eco*RI-Kozak sequence-A0A1G4I464-His—stop codon—*Hind*III
A0A1G4I740	NdeI-ATG-A0A1G4I740-His tag—Stop codon—*Hind*III

**Table 3 vetsci-11-00127-t003:** Results of the indirect ELISA using A0A1G4I8N3 protein (cut-off value: 50% S/P%) compared with the complement fixation test (gold standard). Diagnostic sensibility, specificity and accuracy and confidence limits (95% probability).

		Complement Fixation		
		Positive	Negative	Total
*Indirect ELISA* *A0A1G4I8N3*	Positive	13	37	50
	Negative	2	43	45
	Total	15	80	95
Diagnostic sensitivity		86.7%	C.L. (95%) 53.7–96.0	
Diagnostic specificity		53.8%	C.L. (95%) 42.9–64.3	
Diagnostic accuracy		59.0%	C.L. (95%) 48.9–68.3	

**Table 4 vetsci-11-00127-t004:** Results of the indirect ELISA using the A0A1G4I464 protein (cut-off value: 80% S/P%) compared with the complement fixation test (gold standard). Diagnostic sensibility, specificity and accuracy, and confidence limits (95% probability).

		Complement Fixation		
		Positive	Negative	Total
*Indirect ELISA* *A0A1G4I464*	Positive	8	33	41
	Negative	7	47	54
	Total	15	80	95
Diagnostic sensitivity		53.3%	C.L. (95%) 23.6–75.3	
Diagnostic specificity		58.7%	C.L. (95%) 47.8–68.9	
Diagnostic accuracy		57.9%	C.L. (95%) 47.8–67.3	

**Table 5 vetsci-11-00127-t005:** Results of the indirect ELISA using the A0A1G4I740 protein (cut-off value: 55% S/P%) compared with the complement fixation test (gold standard). Diagnostic sensibility, specificity and accuracy, and confidence limits (95% probability).

		Complement Fixation		
		Positive	Negative	Total
*Indirect ELISA* *A0A1G4I740*	Positive	11	28	39
	Negative	4	52	56
	Total	15	80	95
Diagnostic sensitivity		73.3%	C.L. (95%) 47.6–89.0	
Diagnostic specificity		65.0%	C.L. (95%) 54.0–74.6	
Diagnostic accuracy		66.3%	C.L. (95%) 56.3–75.0	

**Table 6 vetsci-11-00127-t006:** Results of Western blotting using A0A1G4I8N3 protein compared with the complement fixation test (gold standard). Diagnostic sensibility, specificity and accuracy and confidence limits (95% probability).

		Complement Fixation		
		Positive	Negative	Total
*Immunoblotting* *A0A1G4I8N3*	Positive	13	6	19
	Negative	2	74	76
	Total	15	80	95
Diagnostic sensitivity		86.7%	C.L. (95%) 61.7–96.0	
Diagnostic specificity		92.5%	C.L. (95%) 84.6–96.5	
Diagnostic accuracy		91.6%	C.L. (95%) 84.2–95.6	

**Table 7 vetsci-11-00127-t007:** Results of Western blotting using A0A1G4I464 protein compared with the complement fixation test (gold standard). Diagnostic sensibility, specificity and accuracy and confidence limits (95% probability).

		Complement Fixation		
		Positive	Negative	Total
*Immunoblotting* *A0A1G4I464*	Positive	7	15	22
	Negative	8	65	73
	Total	15	80	95
Diagnostic sensitivity		46.7%	C.L. (95%) 24.7–70.1	
Diagnostic specificity		81.3%	C.L. (95%) 71.3–88.3	
Diagnostic accuracy		75.8%	C.L. (95%) 66.3–83.3	

**Table 8 vetsci-11-00127-t008:** Results of Western blotting using A0A1G4I740 protein compared with the complement fixation test (gold standard). Diagnostic sensibility, specificity and accuracy and confidence limits (95% probability).

		Complement Fixation		
		Positive	Negative	Total
*Immunoblotting* *A0A1G4I740*	Positive	12	29	41
	Negative	3	51	54
	Total	15	80	95
Diagnostic sensitivity		80.0%	C.L. (95%) 54.4–92.7	
Diagnostic specificity		63.8%	C.L. (95%) 52.8–73.4	
Diagnostic accuracy		66.3%	C.L. (95%) 56.3–75.0	

## Data Availability

Data is contained within the article.

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
