# Peer review of "Analysis of *Trypanosoma equiperdum* Recombinant Proteins for the Serological Diagnosis of Dourine"

_vetsci, 2024, doi:10.3390/vetsci11030127_

Round 1
Reviewer 1 Report
Comments and Suggestions for Authors
The presented article demonstrates that at least one recombinant protein of T. equiperdum has potential for use as a diagnosis by the method of immunobloting.
However, in my opinion, in addition to the experiments carried out, it is vital to demonstrate that there is no cross-reaction with serums obtained from animals infected with T evansi. Without this proof, the result is fragile. So I suggest that the authors do the test with this serum.
Below are some considerations:
Abstract:
1- A conclusion sentence is missing in the abstract. It ends abruptly with the results.
Introduction:
1- I suggest improving the formatting. Join paragraphs.
Material and methods:
1- What are the asterisks and dots presented in the FASTA sequences of proteins?
2- In table 2, Eco and Hind should be written in italics.
3- Why in the preservation buffer solution of protein A0A1G41740 have arginine and in the buffering solution of the other proteins do not?
4-... performed at IZSAM. Quote the full name at the first appearance in the text.
5- Why were only the proteins expressed in baculoviruses expressed on a large scale?
6- In subsections 2.3 and 2.4, why the differences in BSA concentration, incubation time, dilutions, etc. relative protein A0A1G4I740?
Results:
1- In figure 1,lane 1, what is the band that appears around 9 kDa?
2- In Figure 2, in the legend, this described "horse sera negative" where is this result in the figure?
Discussion:
1- What is the explanation why negative serums that did not recognize the native proteins in previous papers are recognizing the recombinant protein in this paper, both in the ELISA test and in the immunobloting test?
2-I suggest that the authors read the article "Kim J, Álvarez-Rod A, Li Z, Radwanska M, Magez S. Recent Progress in the Detection of Surra, a Neglected Disease Caused by Trypanosoma evansi with a One Health Impact in Large Parts of the Tropic and Sub-Tropic World. Microorganisms. 2024; 12(1):44. https://doi.org/10.3390/microorganisms12010044" to verify specificity, sensitivity and accuracy issues.
Comments on the Quality of English LanguageNo comments
Reviewer 2 Report
Comments and Suggestions for Authors
The paper from Mirella et al seeks to address a topic of veterinary importance (dourine diagnosis) that lacks of studies and with a limited understanding of the pathogenesis of the disease.
Overall, the manuscript is well structured and the topics discussed are presented well, though there are some issues that need to be addressed.
1. Simple summary / abstract - the authors should rewrite the simple summary as it duplicates much of what is written in the abstract. The simple summary does not cite what the proteins are, just uses the acronyms, which is not helpful to lay readers.
2. Page 2, line 28; animals are not patients here.
3. Page 2, line 47. The paper lacks any description of the diseases; i t is important to describe some of the pathology, and what treatments are available (If any, other than culling).
4. Page 2, line 73. The authors could explain more about how they choose the 3 proteins.
5. Page 2 (Methods), Expression and production of recombinant proteins. The authors explained all the process of production of 2 of the protein, but not of the third one in bacteria. Details should be added to methods here.
6. The major issue for the paper is the number of samples positive used and their provenance. The authors are using only 15 positive sera, and collected in 2011; I do not think these are sufficient to test for cross-reactivity and hence quantify a true specificity for the tests. Why haven't the authors collected samples during the past decade? I would not be confident of drawing conclusions on sensitivity and specificity of my test(s) based on 15 samples only.
7. IN Figure 1, there are two gels bands, and this is arguably the best one in the paper; but there is no discussion as to what this band is. It's not the PPIase, so is it a contaminant?
8. The authors do discuss some of the limitations of the study, which I think is to premature with such a small number of positive samples (given the distribution of the disease globally).
Comments on the Quality of English LanguageEnglish is generally fine; there are a few grammatical errors that can be easily addressed for better flow.
Round 2
Reviewer 1 Report
Comments and Suggestions for Authors
The authors addressed the revisor's doubts and made the necessary corrections.
I maintain my recommendation for them to undergo testing using serums from animals infected with T. evansi. This could serve as an incentive for the diagnostic tool in the future.
Reviewer 2 Report
Comments and Suggestions for Authors
The authors have answered my questions and made the additions requested. the paper is now a pilot study as intended, given the low numbers of sera available. This is an interesting pathogen and disease and I look forward to reading more on this in the future